# Meta-Analysis of RNA Sequencing Data of Arabidopsis and Rice under Hypoxia

**DOI:** 10.3390/life12071079

**Published:** 2022-07-19

**Authors:** Keita Tamura, Hidemasa Bono

**Affiliations:** 1Laboratory of Genome Informatics, Graduate School of Integrated Sciences for Life, Hiroshima University, 3-10-23 Kagamiyama, Higashi-Hiroshima 739-0046, Hiroshima, Japan; tamurak@hiroshima-u.ac.jp; 2Laboratory of BioDX, Genome Editing Innovation Center, Hiroshima University, 3-10-23 Kagamiyama, Higashi-Hiroshima 739-0046, Hiroshima, Japan

**Keywords:** meta-analysis, hypoxia, flooding, submergence, waterlogging, Arabidopsis, rice

## Abstract

Hypoxia is an abiotic stress in plants. Flooding resulting from climate change is a major crop threat that increases the risk of hypoxic stress. The molecular mechanisms underlying hypoxia in plants were elucidated in recent years, but new genes related to this stress remain to be discovered. Thus, we aimed to perform a meta-analysis of the RNA sequencing (RNA-Seq) data of Arabidopsis (*Arabidopsis thaliana*) and rice (*Oryza sativa*) under hypoxia. We collected 29 (Arabidopsis) and 26 (rice) pairs of RNA-Seq data involving hypoxic (including submergence) and normoxic (control) treatments and extracted the genes that were commonly upregulated or downregulated in the majority of the experiments. The meta-analysis revealed 40 and 19 commonly upregulated and downregulated genes, respectively, in the two species. Several WRKY transcription factors and cinnamate-4-hydroxylase were commonly upregulated, but their involvement in hypoxia remains unclear. Our meta-analysis identified candidate genes for novel molecular mechanisms in plants under hypoxia.

## 1. Introduction

Flooding is a major abiotic stress for crop production, and water availability extremes (drought and flooding) have become frequent, owing to climate change [1]. Flooding can be classified into waterlogging (partial coverage of water) and submergence (complete coverage of water), both of which limit oxygen availability to plants and lead to hypoxia [2]. Considering the importance of molecular oxygen in eukaryotes, several studies have elucidated and compared the molecular mechanisms by which plants and animals respond to hypoxia [3,4]. In metazoans, hypoxia-inducible factor (HIF) is a transcription factor (TF) that regulates the transcriptional changes induced by oxygen availability [3,5]. In plants, the group VII ethylene response factor TFs (ERFVIIs) induce transcriptional changes under hypoxia [3]. The HIF and ERFVIIs are a functionally analogous pair that stabilizes under hypoxia and degrades under oxygenated conditions; however, their pathways are mechanistically different [3].

The molecular mechanisms of plants under hypoxic conditions, including flooding, submergence, and waterlogging, were extensively studied in Arabidopsis (*Arabidopsis thaliana*) and rice (*Oryza sativa*) [2]. Ethylene is entrapped in plants upon flooding, causing signaling cascades for adaptation to hypoxia [6]. The discovery of ERFVII genes, such as *SUBMERGENCE 1A* (*SUB1A*) [7,8], *SNORKEL1* (*SK1*), and *SNORKEL2* (*SK2*) [9], in rice varieties with submergence tolerance has led to the study of this gene family in Arabidopsis [6]. In Arabidopsis, five ERFVII TFs have been identified [10,11]. Among them, *HYPOXIA RESPONSIVE ERF1* (*HRE1*) and *HRE2* are expressed under hypoxia [12], whereas *RELATED TO APETALA2.12* (*RAP2.12*), *RAP2.2*, and *RAP2.3* are constitutively expressed under normoxia [11]. Arabidopsis has 49 core hypoxia-responsive genes (HRGs), including *HRE1* and *HRE2* [13]. The stabilized ERFVII TFs under hypoxia are involved in the upregulation of over half of the 49 HRGs [14]. The transactivation studies of Arabidopsis ERFVII TFs indicated that *RAP2.12*, *RAP2.2*, and *RAP2.3* act redundantly to activate HRGs, whereas *HRE1* and *HRE2* play minor roles as activators [11]. The stability of the ERFVII TFs, with oxygen-sensing mechanisms or protein–protein interactions with the ERFVII TFs, have been extensively studied [2,6]. However, other molecular mechanisms outside the control of the ERFVII TFs remain to be determined.

Identifying novel molecular mechanisms under specific biological conditions is an important biological study, but hypothesis construction is generally influenced by well-studied theories and publication bias. Thus, a meta-analysis, which is a data-driven and unbiased collective analysis of multiple studies, is a promising strategy to provide novel insights [15,16]. Recently, the number of transcriptome analyses has increased dramatically, and most transcriptome studies have performed RNA sequencing (RNA-Seq) using the Illumina platform (All of Gene Expression (AOE); https://aoe.dbcls.jp, accessed on 16 July 2022) [17]. This abundance of studies makes a meta-analysis of RNA-Seq data feasible, using publicly available data. This strategy can lead to the identification of novel hypoxia-inducible genes in human cell lines or tissue specimens [15].

In this study, we aimed to perform a meta-analysis of public RNA-Seq data of Arabidopsis and rice under hypoxia to identify novel hypoxia-inducible genes in plants. The experiments involving hypoxic treatments, including submergence and waterlogging, showed significant enrichment of the genes with hypoxia-related gene ontology (GO) in the collection of frequently upregulated genes under this stress. The expression of several WRKY TFs and cinnamate-4-hydroxylase (C4H) was commonly upregulated. Our datasets contribute to the functional characterization of novel hypoxia-inducible genes in plants.

## 2. Materials and Methods

### 2.1. Curation of Public Gene Expression Data

To obtain public gene expression data related to hypoxia responses in plants, we searched the Gene Expression Omnibus (GEO) [18] using the following keywords: (((“hypoxia” OR “hypoxic” OR “low oxygen” OR “submergence” OR “submerged” OR “waterlogging”) AND “Arabidopsis thaliana”(porgn:__txid3702)) AND “gse”(Filter) AND “Expression profiling by high throughput sequencing”(Filter)) for Arabidopsis, and (((“hypoxia” OR “hypoxic” OR “low oxygen” OR “submergence” OR “submerged” OR “waterlogging”) AND “Oryza sativa”(porgn:__txid4530)) AND “gse”(Filter) AND “Expression profiling by high throughput sequencing”(Filter)) for rice. Using the list of GSE accession numbers matched with the keywords, we obtained the corresponding SRP accession numbers and metadata with the Python package pysradb (v1.1.0). The obtained metadata were manually curated to focus on RNA-Seq of total mRNA and paired experiments of hypoxic and normoxic treatments. As a result, 29 (Arabidopsis) and 26 (rice) pairs of RNA-Seq data involving hypoxic (including submergence and waterlogging) and normoxic (control) treatments were created for this meta-analysis. The list of the RNA-Seq data used for this meta-analysis is available at figshare files 1a,b [19].

### 2.2. Gene Expression Quantification

The FASTQ-formatted files for each RNA-Seq run accession number were retrieved using the SRA Toolkit (v2.11.0) (https://github.com/ncbi/sra-tools, accessed on 16 July 2022) with prefetch and fasterq-dump commands. The FASTQ-formatted files for the same experiment were concatenated. The quality control of the raw reads was performed using Trim Galore (v0.6.7) (https://github.com/FelixKrueger/TrimGalore, accessed on 16 July 2022) with default settings. The transcript quantification was conducted using Salmon (v1.6.0) [20] against reference cDNA sequences downloaded from Ensembl Plants [21] (Arabidopsis_thaliana.TAIR10.cdna.all.fa.gz for Arabidopsis and Oryza_sativa.IRGSP-1.0. cdna.all.fa.gz for rice) with default settings (“-l A” was specified for automatic library type detection). These processes were performed using a set of scripts available at https://github.com/bonohu/SAQE (accessed on 16 July 2022) [22], with small modifications. The quantitative RNA-Seq data, which were calculated as transcripts per million (TPM), are available at figshare (figshare files 2a,b) [19].

### 2.3. Calculation of HN-Ratio and HN-Score

To normalize the gene expression between the different studies, the change in gene expression levels between each hypoxia and normoxia pair was measured as the HN-ratio [15]. The HN-ratio *R* was calculated using the following equation:(1)$$R=\frac{T_{hypoxia}+1}{T_{normoxia}+1}$$
where *T_hypoxia_* and *T_normoxia_* are the TPM values of each gene in the hypoxic and normoxic treatments, respectively.

The HN-score [15] of each transcript was calculated to evaluate the changes in the gene expression under hypoxia across the experiments. We treated the biological replicates with the same conditions as the individual experiment. In this study, the transcripts with an HN-ratio of two or greater were considered to be upregulated, whereas those with an HN-ratio of 0.5 or less were considered to be downregulated. The HN-score of each transcript was calculated by subtracting the number of experiments with downregulated expression from the number of experiments with upregulated expression. The HN-ratio and HN-score were calculated using a set of scripts on https://github.com/no85j/hypoxia_code (accessed on 16 July 2022) [15]. The HN-ratio and HN-score are available at figshare (figshare files 3a,b, and 4a,b) [19].

### 2.4. Functional Annotation of Transcripts

For the Arabidopsis data, the “Gene name” and “Gene description” for each transcript were obtained from Ensembl Biomart (Arabidopsis thaliana genes (TAIR10)) [23]. To find the orthologs in humans, we searched each transcript against the human proteins (Homo_sapiens.GRCh38.pep.all.fa.gz) with an E-value cutoff of 1e-5 by blastx (v2.12.0) program, and obtained the “Gene name” and “Gene description” (Human (GRCh38.p13)) for the top hits.

For the rice data, the “Gene name” and “Gene description” for each transcript were obtained from Ensembl Biomart (Oryza sativa Japonica Group genes (IRGSP-1.0)). To find the orthologs in Arabidopsis, we searched each transcript against the Arabidopsis proteins (Arabidopsis_thaliana.TAIR10.pep.all.fa.gz) with an E-value cutoff of 1e-10 by blastx (v2.12.0), and obtained the “Gene name” and “Gene description” (Arabidopsis thaliana genes (TAIR10)) for the top hits. Similarly, the orthologs in humans were obtained as described in the Arabidopsis data.

### 2.5. Gene Set Enrichment Analysis

The gene set enrichment analysis was performed using Metascape (https://metascape.org/, accessed on 16 July 2022) [24], with the default settings (enrichment analysis with GO Biological Processes, KEGG Pathway and WikiPathways; *p*-value < 0.01, a minimum count of three, and an enrichment factor >1.5 (the enrichment factor is the ratio between the observed counts and the counts expected by chance)), or modified the ontology sources to GO Molecular Functions or GO Cellular Components. All of the genes in the genome were used as the enrichment background, and up to the top 20 enriched clusters were visualized. The rice transcripts were analyzed using orthologous transcript IDs in Arabidopsis. The Venn diagrams were constructed using a web tool (https://bioinformatics.psb.ugent.be/webtools/Venn/, accessed on 16 July 2022).

### 2.6. Protein–Protein Interaction Analysis

The protein–protein interaction was analyzed and visualized using STRING (v11.5) (https://string-db.org/, accessed on 16 July 2022) [25], with default settings.

## 3. Results

### 3.1. Curation of RNA-Seq Data for Meta-Analysis

Considering that hypoxia in plants was associated with submergence and waterlogging, we used “submergence,” “waterlogging,” and “hypoxia” as the keywords in the GEO search. The searches were refined in the RNA-Seq data because, unlike the microarray data with various platforms, most of the RNA-Seq data are based on the Illumina platform, rendering them suitable for comparative analyses between studies. We collected 29 (Arabidopsis) and 26 (rice) pairs of RNA-Seq data of hypoxic and normoxic treatments (figshare files 1a,b) [19]. The data for Arabidopsis included hypoxic and submergence treatments. Although the experiments with some mutants or transgenic lines were included, all of the data for Arabidopsis were derived with Col-0 background plants. A total of 21 of the 29 pairs of the RNA-Seq data were sampled from seedlings, and the remaining eight pieces of data were from leaves (figshare file 1a) [19]. The rice data included submergence or waterlogging treatments. M202(Sub1) [8], Varshadhan, and Rashpanjor [26] are flood-tolerant cultivars, whereas the majority of the experiments used flood-intolerant cultivars. A total of 12 of the 26 pairs of the RNA-Seq data were sampled from the above ground parts, and the remaining 14 pieces of data were from the roots (figshare file 1b) [19].

### 3.2. Characteristics of Upregulated Transcripts under Hypoxia

The meta-analysis of the RNA-Seq data of Arabidopsis and rice under hypoxia was performed by calculating the HN-score and HN-ratio of all of the transcripts [15]. The HN-score of each transcript is the difference between the number of experiments with downregulated expression and the number of experiments with upregulated expression; therefore, the higher scores indicate global trends of upregulation across the experiments, whereas the lower (minus) scores indicate global trends of downregulation across the experiments. We defined the top 1% transcripts with the highest and lowest HN-scores in the meta-analysis as upregulated and downregulated, respectively (figshare files 5a–d) [19]. The HN-score ranges of the upregulated and downregulated transcripts are summarized in Table 1.

The gene set enrichment analysis of the upregulated transcripts in Arabidopsis and rice showed the enrichment of the hypoxia-related terms in both of the species (Figure 1a,b). In Arabidopsis, the most significantly enriched term was GO:0071456 (cellular response to hypoxia) (Figure 1a). This was also the same when we analyzed the hypoxic treatments and submergence treatments separately (figshare file 7) [19]. The highest HN-score of 26 was observed in AT4G24110.1, AT5G47910.1 (*RBOHD*), and AT1G76650.3 (*CML38*), all of which were included in the term GO:0071456 (figshare file 5a) [19]. AT4G24110, known as *HYPOXIA-RESPONSIVE UNKNOWN PROTEIN 40* (*HUP40*), is a member of 16 HUP genes in Arabidopsis [27]. The RESPIRATORY BURST OXIDASE HOMOLOGUE D (RBOHD) is a NADPH oxidase involved in hydrogen peroxide (H_2_O_2_) production under hypoxia [28], and in plant immunity [29]. The CALMODULIN-LIKE 38 (CML38) is a core hypoxia response calcium sensor protein [30]. In rice, the most significantly enriched term was GO:0001666 (response to hypoxia), the parent term of GO:0071456 (https://www.ebi.ac.uk/QuickGO/term/GO:0001666, accessed on 16 July 2022). The highest HN-score of 21 was observed in Os06t0605900-01 (*OsFbox316*) and Os01t0129600-00, of which only Os01t0129600-00 was included in the term GO:0001666 (figshare file 5b) [19]. The OsFbox316 ortholog in maize is upregulated under waterlogging [31]. The Os01g0129600 (Os01t0129600-00) is similar to LOB DOMAIN-CONTAINING PROTEIN 40 (LBD40). The Arabidopsis *LBD40* expression, together with *LBD4* and *LBD41* expression, is upregulated under hypoxia [32]. We also performed the gene set enrichment analysis against GO Molecular Function and GO Cellular Component (figshare file 8) [19]. Unlike the GO Biological Process, the most significantly enriched term did not match between Arabidopsis and rice.

The meta-analysis showed that the representative hypoxia-related genes in Arabidopsis were classified in accordance with previous studies (Table 2). Among the five Arabidopsis ERFVII TFs, the expression of *HRE1* and *HRE2* was upregulated, whereas *RAP2.12*, *RAP2.2*, and *RAP2.3* expression was unchanged (Table 2). All of the Arabidopsis core HRGs (*LBD41*, *PCO1*, *PCO2*, *ADH1*, and *PDC1*), regulated by *RAP2.2* and *RAP2.12* [11], were also upregulated (Table 2). The PLANT CYSTEINE OXIDASEs (PCO1 and PCO2) are involved in the stability of the ERFVII TFs [33], and ALCOHOL DEHYDROGENASE 1 (ADH1) and PYRUVATE DECARBOXYLASE 1 (PDC1) are involved in the fermentative metabolism [6]. The expression of rice ERFVII TFs *SUB1B* and *SUB1C* was also upregulated (figshare file 5b) [19]. Considering that most of the experiments on rice involved the cultivar Nipponbare, and that the Nipponbare transcriptome was used as a reference, we excluded the submergence-tolerant ERFVII TFs (*SUB1A*, *SK1*, and *SK2*), which were absent in the Nipponbare genome [7,9], from our meta-analysis.

### 3.3. Characteristics of Downregulated Transcripts under Hypoxia

The gene set enrichment analysis of the downregulated transcripts revealed different trends between Arabidopsis and rice (Figure 2a,b). In Arabidopsis, the most significantly enriched term was GO:0009642 (response to light intensity) (Figure 2a). The darkness is often associated with natural submergence [6]; some of the experiments collected for the meta-analysis of Arabidopsis were conducted under darkness and submergence (figshare file 1a) [19]. These conditions can downregulate the genes related to response to light intensity. However, when we analyzed the hypoxic treatments only, the most significantly enriched term was GO:0016049 (cell growth) (figshare file 7) [19], suggesting that the presence of water or darkness also affects the change of gene expression. The lowest HN-score of −21 was observed in AT5G65730.1 (XTH6) (figshare file 5c) [19]. Some members of the XYLOGLUCAN ENDOTRANSGLUCOSYLASE/HYDROLASE (XTH) gene family have been studied in relation to cell wall expansion [34,35]. In rice, the most significantly enriched term was GO:0006790 (sulfur compound metabolic process) (Figure 2b). The ethylene decreases the antioxidant metallothionein, a small cysteine-rich protein [6,36]. The downregulated genes in the term GO:0006790 may be related to a decrease in metallothionein. The lowest HN-score of −22 was observed in Os05t0217700-01 (OsBURP07) and the two isoforms of Os08g0531000 (Os08t0531000-01 and Os08t0531000-02) (figshare file 5d) [19]. OsBURP07 is downregulated under abscisic acid (ABA) treatment [37]; however, this phenomenon cannot be attributed to the ABA treatment, because ethylene decreases the ABA levels in submerged tissues [38]. The Os08g0531000, also known as NUCLEOTIDE PYROPHOSPHATASE/PHOSPHODIESTERASE 1, negatively regulates starch accumulation and growth [39]. We also performed the gene set enrichment analysis against the GO Molecular Function and GO Cellular Component (figshare file 9) [19]. In the GO Molecular Function, GO:0046906 (tetrapyrrole binding) was commonly enriched in Arabidopsis and rice. This member includes many of the cytochrome P450 monooxygenases, which play an important role in metabolic diversification [40]. The enrichment of this GO may be related to the decreased activity of metabolic processes.

### 3.4. Identification of Commonly Upregulated or Downregulated Genes in Arabidopsis and Rice

To identify the novel candidate genes related to hypoxia in plants, we focused on the genes commonly upregulated or downregulated in Arabidopsis and rice, because they are expected to be related to common mechanisms in plants. The upregulated or downregulated transcript IDs, converted to orthologs in Arabidopsis, were summarized as Arabidopsis gene IDs, and the overlaps between the two species were visualized (Figure 3a,b). We identified 40 and 19 commonly upregulated and downregulated genes, respectively (Table 3 and Table 4, respectively). The gene set enrichment analysis of the overlapping genes showed that the most significantly enriched term of the upregulated genes was GO:0071456 (cellular response to hypoxia), whereas the downregulated genes showed a less confident enriched term (Figure 3c,d).

Among the 40 commonly upregulated genes, 23 were not included in the term GO:0071456, suggesting that these may be novel genes upregulated by hypoxia (Table 3). Notably, three of the four WRKY TFs that were commonly upregulated were not included in the term GO:0071456. The CYP73A5 is C4H, which is a key enzyme in the phenylpropanoid pathway [41]. The *ACC OXIDASE 1* (*ACO1*) does not include the term GO:0071456, but the upregulation of *ACO1* expression under hypoxia has been reported in Arabidopsis [42]. Among the 19 commonly downregulated genes were aquaporin genes *PLASMA MEMBRANE INTRINSIC PROTEIN 2;7* (*PIP2*;7) and *PIP1;5* (Table 4). PIP2;7 is an active water channel in Arabidopsis [43]; hence, the downregulation of aquaporin genes may be related to the inhibition of excess water transport under submergence. We also analyzed the possible protein–protein interactions of the commonly upregulated and downregulated genes, using STRING (figshare file 10) [19]. Although most of the commonly upregulated genes were in association with co-expression and text mining evidence, none of the known or predicted interactions was observed.

## 4. Discussion

In this study, we collected 29 (Arabidopsis) and 26 (rice) pairs of the RNA-Seq data of hypoxic and normoxic treatments and performed a meta-analysis of the changes in gene expression under hypoxia. We treated the biological replicates from the same series of experiments as an individual experiment, and confirmed the swapping of the pairs in the replicates produced similar results in the gene set enrichment analysis (figshare file 11) [19]. Although the present meta-analysis included fewer experiments than a similar meta-analysis performed in humans [15], it clearly showed the enrichment of the hypoxia-related GO terms in Arabidopsis and rice (Figure 1a,b). This result suggests that the present meta-analysis reflects the global trends in differential gene expression under hypoxia. Moreover, the representative hypoxia-related genes in Arabidopsis showed the same expression patterns as those in previous studies (Table 2). We believe that the dataset used in our meta-analysis correctly reflects our scientific knowledge of gene expression under hypoxia.

Our meta-analysis aimed to elucidate the unknown molecular mechanisms. To this end, we focused on the commonly upregulated or downregulated genes in Arabidopsis and rice. We identified 40 upregulated and 19 downregulated genes in both of the species (Table 3 and Table 4). Among the 40 commonly upregulated genes, four WRKY TFs were included, of which three were not included in the hypoxia-related GO terms (Table 3). The role of WRKY TFs in the hypoxia response has not been studied in Arabidopsis; however, in persimmon (*Diospyros kaki*), the expression of some WRKY TFs, including *DkWRKY1*, is upregulated by hypoxia and the process involves *DkPDC2* transactivation [44]. Another report showed that rice *WRKY62* is upregulated under hypoxia and activates hypoxia genes, while repressing the defense-related diterpenoid phytoalexin factor under hypoxia [45]. However, in the present meta-analysis, *WRKY62* (Os09t0417800-01 and Os09t0417800-02) expression was not upregulated. Determining the role of WRKY TFs, including the four commonly upregulated genes, in the response to hypoxia, is a valuable research goal. We also detected an upregulation in the expression of *CYP73A5* (*C4H*), a key enzyme of the phenylpropanoid pathway, in Arabidopsis and rice, and its corresponding gene was not included in the hypoxia-related GO term (Table 3). However, the upstream gene (phenylalanine ammonia-lyase (*PAL*)) and the downstream gene (4-coumaroyl CoA ligase (*4CL*)) of C4H in the general phenylpropanoid pathway were not upregulated in either Arabidopsis or rice (figshare files 6a,b) [19]. A previous study suggested that flavonoid and lignin biosyntheses are suppressed in Arabidopsis under hypoxia [31]. The effect of *C4H* upregulation under hypoxia may be the subject of future studies.

Finally, we compared our results with those of a previous meta-analysis of hypoxia in human cell lines and tissue specimens [15]. In our meta-analysis, two well-studied HRGs, namely, *ADH1* and *PCO2*, were upregulated in both Arabidopsis and rice (Table 3). The orthologs in the humans that were identified in this study are ADH5 and ADO, respectively (figshare file 5a) [19]. A previous meta-analysis of hypoxia in humans obtained HN-scores (1.5-fold threshold; 495 experimental pairs) of −34 and −84 for ADH5 and ADO, respectively [46], indicating that their orthologs in humans are downregulated under hypoxia. The fermentative metabolism of pyruvate is directed toward lactate in animals via lactate dehydrogenase (LDH), in contrast to the ethanol formation in plants via PDC and ADH [26]. Indeed, the HN-score of LDHA in a previous meta-analysis in humans was 334, wherein it ranked in the top 100 upregulated genes [47]. ADO is an enzymatic oxygen sensor with an equivalent role in plant PCOs, identified as a component of an alternative mechanism for oxygen-sensitive proteolysis in mammals [3,48]. Our meta-analysis suggests distinct molecular mechanisms under hypoxia in plants and animals.

## Figures and Tables

**Figure 1 life-12-01079-f001:**
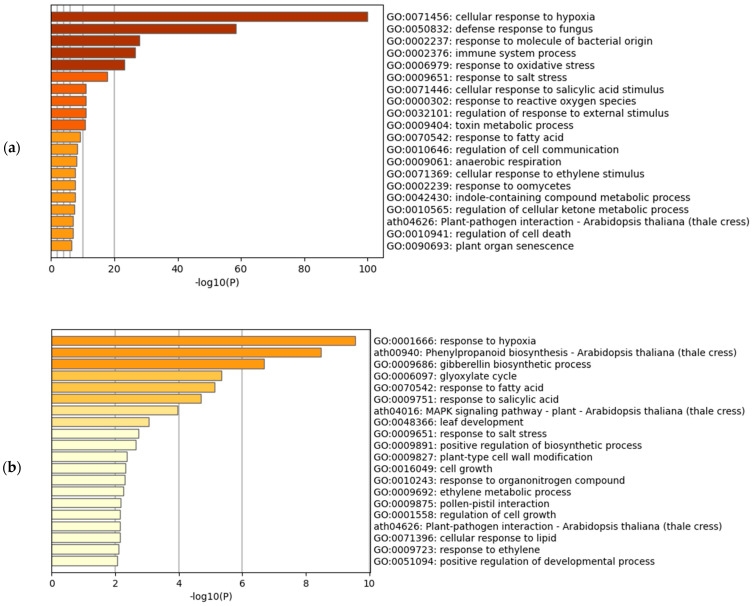
Gene set enrichment analysis of hypoxia-inducible upregulated genes. (**a**) Meta-analysis of Arabidopsis; (**b**) Meta-analysis of rice using the corresponding Arabidopsis genes.

**Figure 2 life-12-01079-f002:**
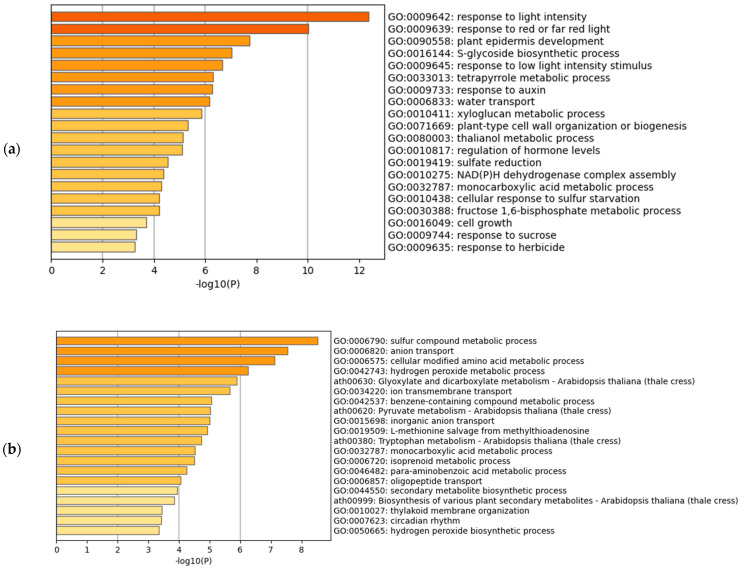
Gene set enrichment analysis of hypoxia-inducible downregulated genes. (**a**) Meta-analysis of Arabidopsis; (**b**) Meta-analysis of rice using the corresponding Arabidopsis genes.

**Figure 3 life-12-01079-f003:**
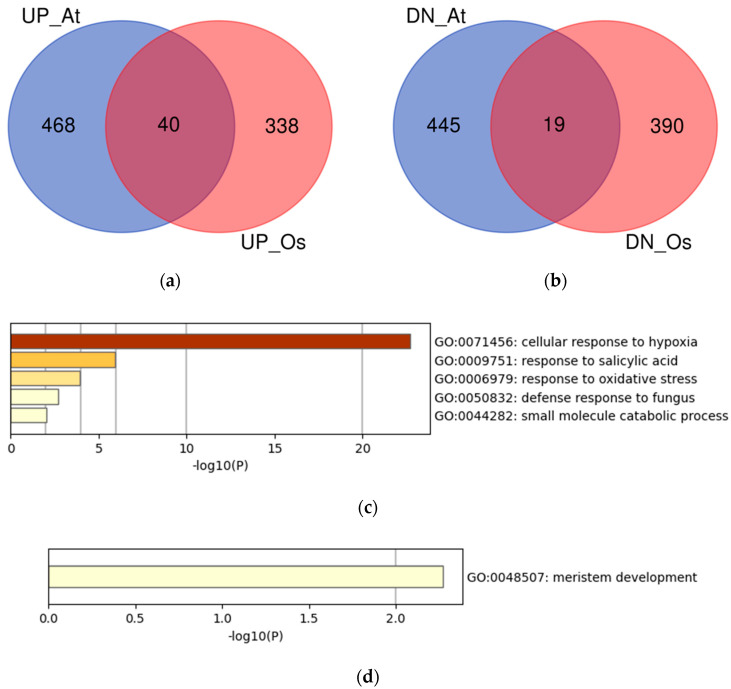
Analysis of the overlaps of upregulated or downregulated genes in Arabidopsis and rice using the Arabidopsis gene IDs. (**a**) Venn diagram of the upregulated genes in Arabidopsis (UP_At) and rice (UP_Os); (**b**) Venn diagram of the downregulated genes in Arabidopsis (DN_At) and rice (DN_Os); (**c**) Gene set enrichment analysis of the commonly upregulated genes; (**d**) Gene set enrichment analysis of the commonly downregulated genes.

**Table 1 life-12-01079-t001:** HN-scores of hypoxia-inducible transcripts identified in the meta-analysis.

Species	Up or Down	HN-Score (*S*)	No. of Transcripts
Arabidopsis	upregulated	11 ≤ *S* ≤ 26	561
Arabidopsis	downregulated	−21 ≤ *S* ≤ −9	493
Rice	upregulated	11 ≤ *S* ≤ 21	606 (477) ^1^
Rice	downregulated	−22 ≤ *S* ≤ −10	570 (489) ^1^

^1^ Numbers in the parentheses are the number of transcripts annotated to Arabidopsis orthologs.

**Table 2 life-12-01079-t002:** HN-scores of representative transcripts related to hypoxia response in Arabidopsis.

Gene ID	Gene Name	Meta-Analysis ^1^
AT1G72360	HRE1	Up
AT2G47520	HRE2	Up
AT1G53910	RAP2.12	–
AT3G14230	RAP2.2	–
AT3G16770	RAP2.3	–
AT3G02550	LBD41	Up
AT5G15120	PCO1	Up
AT5G39890	PCO2	Up
AT1G77120	ADH1	Up
AT4G33070	PDC1	Up

^1^ Up, upregulated; Down, downregulated; –, unchanged. Results are summarized as genes, as each isoform of the genes in the table showed the same results.

**Table 3 life-12-01079-t003:** Commonly upregulated transcripts in Arabidopsis and rice, indicated as Arabidopsis gene IDs.

Gene ID	Gene Name	In GO:0071456 ^1^	Gene ID	Gene Name	In GO:0071456 ^1^
AT1G80840	WRKY40	–	AT2G46400	WRKY46	Yes
AT1G77120	ADH1	Yes	AT3G11820	SYP121	–
AT5G24530	DMR6	Yes	AT5G39890	PCO2	Yes
AT4G20860	FAD-OXR	Yes	AT3G11930	–	–
AT3G22060	CRRSP38	–	AT2G23810	TET8	–
AT1G18390	–	–	AT4G10265	–	Yes
AT1G15670	–	–	AT5G06320	NHL3	Yes
AT1G72210	BHLH96	–	AT1G43800	S-ACP-DES6	Yes
AT3G61060	AtPP2-A13	–	AT2G38470	WRKY33	–
AT4G33050	EDA39	–	AT5G50200	NRT3.1	–
AT5G47060	–	Yes	AT3G29970	–	–
AT4G10270	–	–	AT3G02550	LBD41	Yes
AT1G72360	HRE1	Yes	AT2G30490	CYP73A5	–
AT4G23810	WRKY53	–	AT4G27450	–	Yes
AT2G47520	ERF071	Yes	AT1G68840	RAV2	Yes
AT1G14870	PCR2	–	AT2G19590	ACO1	–
AT5G41080	GDPD2	Yes	AT2G40000	HSPRO2	Yes
AT5G48540	CRRSP55	–	AT2G36690		
AT1G25560	TEM1	Yes	AT5G06570	–	–
AT5G25930	–	–	AT5G64120	PER71	–

^1^ GO:0071456 (cellular response to hypoxia).

**Table 4 life-12-01079-t004:** Commonly downregulated transcripts in Arabidopsis and rice, indicated as Arabidopsis gene IDs.

Gene ID	Gene Name	Gene ID	Gene Name	Gene ID	Gene Name
AT4G35100	PIP2;7	AT2G39890	PROT1	AT5G50740	–
AT3G45780	PHOT1	AT2G29630	THIC	AT5G57180	CIA2
AT2G38760	ANN3	AT1G28570	–	AT4G23400	PIP1;5
AT4G18570	–	AT1G16880	ACR11	AT3G55760	–
AT4G19170	CCD4	AT5G42510	DIR1	AT1G62180	APR2
AT1G20020	LFNR2	AT3G18000	NMT1		
AT5G11420	–	AT5G05980	FPGS1		

## Data Availability

The data presented in this study are openly available in figshare [19].

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
