# Peer review of "Meta-Analysis of RNA Sequencing Data of Arabidopsis and Rice under Hypoxia"

_life, 2022, doi:10.3390/life12071079_

Round 1
Reviewer 1 Report
Tamura, K; Bono, H, 2022 (in review): Meta-analysis of RNA Sequencing Data of Arabidopsis and Rice under Hypoxia.
General note
The authors designed a good study however the analyses performed, and results presented do not sufficiently address the main questions arising from the study.
Major comments
- There are many technical aspects of assessing and comparing expression profiles between samples of different organs and species collected by different studies. What tissues were represented in this study? What steps did you take to address the different library construction protocols that could bias your meta-analysis?
- Under normaxia, were the tissues from which RNA was extracted and sequenced corresponding to the same tissues under hypoxia?
- Please provide a table summary / list of all tissues by species and studies that were used in this meta-analysis in the methods and materials section
- How did you normalize between the different studies?
- Did you consider checking for how the different candidate genes interact using PPI network?
- Apart from biological processes, what other GO Terms did you perform enrichment on? Please add these to the results and discussion of your work
Minor comments
- L 13: This is not an entirely accurate statement. Please revise.
- L 117-120: why is this step necessary?
- L 92, 94: What settings and/or parameters were used?
- L 128-132: What parameters and/or settings did you use for enrichment analysis? Was enrichment scored over whole genome or how did you calculate this? Why?
Reviewer 2 Report
The meta-analysis of RNA-seq experimental data is crucial for re-using and rediscovering our predecessors' efforts. On the other hand, it is not easy to compare the results of experimental groups that are not precisely the same. In this study, we compared the results of experiments with simple indices based on the comparison of up-regulation and down-regulation. The results reveal a common molecular mechanism under low oxygen conditions in Arabidopsis and rice, indicating that this method is effective as a meta-analysis.Here are some comments :
1. In line 85, "Manually curated," would you like to describe specifically from what perspective the data set was selected?
2. In line 85, it is difficult to determine whether it is OK to extract each "Biological replicate" and make it into an arbitrary one-to-one combination. Would other combinations of treatment and control produce the same results?
3. In lines 87 and figshare 1a and b, the table is prepared based on the combination calculated by TPM. However, should the RUN accessions from the same experiment be put together in one line? Also, would you like to state clearly that columns B and C are the RUN accessions?
4. In lines 103 to 104, with a single quote " 1' " is attached to the denominator of the formula for the HN-ratio. Is this a misprint?
5. In Lines 141 to 142: Is there any difference in up/down regulation between hypoxia treatment and waterlogging in Arabidopsis thaliana? Is it possible that the regulation of genes is altered in the presence/absence of water? Is it possible that Arabidopsis thaliana responds differently to waterlogging treatment because it is reoxygenated after hypoxia treatment?
6. In lines 191 to 193, the analysis was performed by excluding genes related to water resistance should be explained in more detail.
